# Synergistic Effects of Environmental Relative Humidity and Initial Water Content of Recycled Concrete Aggregate on the Improvement in Properties via Carbonation Reactions

**DOI:** 10.3390/ma16155251

**Published:** 2023-07-26

**Authors:** Linjian Wu, Wenxiao Zhang, Han Jiang, Xueli Ju, Li Guan, Haicheng Liu, Songgui Chen

**Affiliations:** 1National Engineering Research Center for Inland Waterway Regulation, School of River and Ocean Engineering, Chongqing Jiaotong University, 66 Xuefu Road, Nan’an District, Chongqing 400074, China; wljabgf@126.com (L.W.); 15213118329@163.com (H.J.); juxueli97@126.com (X.J.); 2Sichuan Communication Surveying & Design Institute Co., Ltd., Taisheng Bei Road, Qingyang District, Chengdu 610017, China; guanli9798@126.com; 3Tianjin Research Institute for Water Transport Engineering M.O.T., Tianjin 300456, China; chensg@tiwte.ac.cn

**Keywords:** recycled concrete aggregate (RCA), carbonation modification effect, relative humidity (RH), initial water content (IWC), synergistic effect

## Abstract

Moisture is the basis of CO_2_ transport and carbonation reactions in the internal pores of cement-based materials. Too much or too little moisture influences the effect of the carbonation modification of CO_2_ on recycled concrete aggregate (RCA). During the carbonation reaction process of RCA, moisture is mainly derived from the environmental relative humidity (RH) and the initial water content (IWC) of the RCA itself. According to the available literature, most of the studies on the effect of moisture on the carbonation modification of RCA considered either RH or IWC. Further investigations of the synergistic effects of RH and IWC on the improvement in the properties of carbonated recycled concrete aggregate (CRCA) are needed. In this study, accelerated carbonation experiments were conducted for RCA samples with different IWCs under different environmental RHs. The results showed that the best moisture conditions for CRCA property improvement were confirmed as RH = 70% for the dry-state IWC and RH = 50% for the saturated-state IWC. When the RCAs were carbonized under the conditions of high RH with low IWC and low RH with high IWC, CO_2_ had good abilities to permeate and diffuse, with the improvement in CRCA properties achieving excellent levels of performance.

## 1. Introduction

As the most widely used building material in the world, concrete emits a large amount of CO_2_ gas during its production and preparation, which accounts for approximately 5~8% of total human CO_2_ emissions [1]. With the continuous development of China’s urbanization, China’s annual consumption of concrete is very large. The average annual consumption of cement in China accounts for approximately 55% of the world’s total consumption [2], and the average annual output of aggregate accounts for approximately 38% of the world’s total output [3]. The very great demand for concrete will lead to not only an increasing shortage of raw materials such as sand and gravel [4,5], but also an increasing amount of waste concrete. According to the relevant literature [6], China produces approximately 2.36 billion tons of building demolition waste every year, approximately 50~60% of which is concrete waste. At present, the treatment of waste concrete is mostly based on stacking and landfilling [7], which has caused severe adverse effects on the surrounding ecology and environment. To achieve sustainable development and reduce CO_2_ emissions, there is an urgent need to promote the utilization of waste concrete.

Waste concrete is made into recycled concrete aggregate to realize recycling, which is considered to be one of the most effective measures to solve the problem of waste concrete [8,9]. However, in contrast to natural aggregate (NA), recycled concrete aggregate (RCA) is produced by crushing and screening waste concrete that inevitably adheres to old mortar, and the old interfacial transition zone (OITZ) [10] makes the mechanical strength and durability of RCA worse than those of natural aggregate. It has been noted in the literature [11] that the water absorption of RCA was 3~12% greater than that of natural aggregates. The crushing values of RCAs with particle sizes of 4.75~10 mm and 10~20 mm were 45% and 33% larger than those of natural aggregates with the same particle size. At present, the use of the CO_2_-accelerated carbonation method has been proven to be an effective way to improve the physical and mechanical properties of RCAs [12,13,14,15]. RCAs with attached old mortar and OITZs contain Ca(OH)_2_, hydrated calcium silicate (C-S-H), and other components. CO_2_ can react with these components to form dense CaCO_3_ [16] and silica gel (SiO_2_·nH_2_O) [17,18,19] to form reactants that increase the volume of the solid phase by approximately 11~12% [20,21]. Carbonation products can densify the pores and fissures in old mortar and OITZs, improve their microstructure, and enhance the properties of RCA [22,23,24]. After the full carbonation of CO_2_, the porosity of RCA samples could be reduced from 47.6% to 37.2% [25], the crushing index value of carbonated recycled concrete aggregate (CRCA) was 5.9% less than that of RCA [26], the adhesion between mortar and natural aggregate in CRCA was stronger than that of noncarbonized RCA [27], and the microhardness levels of ITZs and attached mortar were increased [28,29]. In addition, the use of CO_2_ to accelerate the carbonation of RCA can create considerable environmental benefits [30].

Moisture is the basis of the carbonation reaction of cement-based materials. Too much or too little moisture affects the dissolution and diffusion of CO_2_ in the pores of RCA [31]. Additionally, moisture participates in the dissolution of solid calcium ions in RCA [32]. Therefore, moisture is closely related to the effect of carbonation on the modification of RCA. As one of the main sources of moisture in the carbonation reaction of RCA, the relative humidity (RH) has an important influence on the degree of the modification of RCA via carbonation. Pu et al. [33] summarized the influencing factors of RCA carbonation modification (including the relative humidity, gas pressure, CO_2_ concentration, temperature, and carbonation duration) and pointed out the optimal relative humidity range for carbonation to treat RCA was 50~70%. Elsalamawy et al. [34] found that the carbonation depth of concrete specimens increased with increasing RH and peaked at RH = 65%. Galan et al. [35] found that after 350 days of carbonation, the mass gain of cement paste samples was the largest when the environmental RH = 53%. Galan et al. [36] also found that the continuity and thickness of the CaCO_3_ layer formed on the surface of Ca(OH)_2_ crystals via the carbonation reaction were related to RH. Gholizadeh-Vayghan et al. [37] observed that the carbonization of RCA was complete under humid conditions (RH > 95%), and its water absorption and porosity did not decrease significantly. The initial water content (IWC) of RCA is another main source of moisture during the carbonation reaction of RCA, and it has a significant effect on the improvement in CRCA properties. Materials with high initial free water contents are more permeable to CO_2_ gas, and more CO_2_ enters the solid phase of RCA [32]. Zhan et al. [38] found that after the carbonation of RCA at IWC = 0.08%, 3.37%, and 5.03%, the carbonation ratio was the largest for the RCA sample with IWC = 3.37%. Pan et al. [39] found that RCA had the smallest porosity after carbonation at IWC = 5%. Thiery et al. [40] observed that when the particle size of RCA was less than 2 mm and the liquid water saturation of the aggregate was less than 0.4, the CO_2_ absorption rate of the RCA sample increased. In summary, recent research on the effect of moisture on the carbonation modification of RCA focused on either environmental RH or the IWC of RCA, but not both. Few related research results were found in the public domain, and they contained conclusions obtained mostly from qualitative analyses. The comprehensive effects of RH and IWC on the modification of RCA via carbonation have not been reported in the literature, and the synergistic effect of RH and IWC on the improvement in the properties of CRCA deserves further study.

Therefore, in this paper, RCA samples in different IWC states were obtained by preparing concrete specimens with determined strength grades, crushing, screening, and pretreatment. Carbonation experiments were performed using RCA samples in different IWC states under three different RH environments. The test results for the water content, actual mass increase rate, carbonation ratio, apparent density, as well as water absorption for the samples of RCA before and after carbonation under different conditions were combined with the electron microscopy scanning results of representative samples, and the comprehensive effect of moisture on the property improvement degree of RCA was explored.

## 2. Experiment and Methods

### 2.1. RCA Production and Preparation

The original concrete was prepared by using PC.42.5R composite Portland cement with a dry density of 3.1 × 10^3^ kg/m^3^, natural coarse aggregate with a nominal particle size of 5~20 mm (apparent density of 2715 kg/m^3^), natural river sand with a fineness modulus of 2.58, and tap water as raw materials. The particle size distribution of fine and coarse aggregate is shown in Figure 1. The mix proportions of concrete were determined according to Chinese standard JGJ55-2011 [41], as shown in Table 1. The concrete cube specimens (100 × 100 × 100 mm^3^) were placed in a standard curing box at a temperature of 20 ± 5 °C and a relative humidity of 90% for 1 day during the curing process. After the initial curing stage, the specimens were soaked in saturated Ca(OH)_2_ solution for 28 days, and the compressive strength of the concrete specimens was measured (as shown in Table 1). The original RCA samples were prepared by crushing and screening the concrete specimens using an E-type gravel crusher and a shock-type standard vibrating screen. RCA samples with their nominal particle size ranges of 10–20 mm were selected for this experiment. According to the method specified in Chinese standard GB/T14685-2022 [42], the initial apparent density of an RCA sample was measured to be 2479 kg/m^3^, and the initial water absorption was 5.34%.

### 2.2. Sample Pretreatment and Carbonation Environment

In this experiment, RCA samples in three different IWC states, i.e., dry, natural, and saturated states (see Figure 2), were prepared using different pretreatment methods. The dry-state RCA sample was prepared by drying the original sample in a blast-drying oven at an ambient temperature of 105 °C to a constant weight. The saturated state RCA sample was prepared by completely soaking the original sample in water for 24 h. The natural-state RCA sample was the original RCA sample without any pretreatment. The RCA-accelerated carbonation experiment was performed in three different carbonation environments, i.e., RH = 50%, 70%, and 90%; the ambient temperature was set to 20 °C; and the CO_2_ concentration was set to 20%. The specific experimental conditions and measured IWC values of the RCA samples are shown in Table 2.

### 2.3. CO_2_ Curing Treatment for RCA

In this RCA-accelerated carbonation experiment, three parallel samples were prepared for each group in Table 2 for a total of 9 groups of 27 RCA samples, and the initial mass of each RCA sample before carbonation was 500 g. For the carbonation experiment, the selected instrument was a TH-W concrete carbonation box that could adjust the carbonation environment parameters in real time. During the accelerated carbonation of RCA samples, the quality changes under different carbonation times were monitored. After a certain interval of time, the quality of each sample was tested and recorded. In the initial stage of carbonation, the quality of the sample was tested frequently. When the quality of the sample remained basically unchanged, the carbonation experiment was completed.

### 2.4. Determination of the Aggregate Properties

#### 2.4.1. Water Content

The water content of the RCA samples before and after carbonation was measured in this experiment, which indicated the transport of moisture during the RCA carbonation reaction. The water content was calculated using Equation (1):(1)$$\omega =\frac{m_{n}-m_{d}}{m_{d}}\times 100\%$$
where *ω* means the water content of the aggregate (%), *m_n_* denotes the mass of the aggregate sample before drying (g), and *m_d_* is the mass of aggregate after completely drying (g).

#### 2.4.2. Apparent Density

The apparent density of the RCA samples before and after carbonation was measured according to the wide-mouth bottle method specified in GB/T14685-2022. The expression for the calculation of the apparent density is shown in Equation (2):(2)$$\rho_{a}=\left(\frac{m_{d}}{m_{d}+m_{t1}-m_{t2}}\right)\times \rho_{w}$$
where *ρ_a_* is the apparent density of the aggregate (kg/m^3^); *m_d_* is the mass of the aggregate after drying (g); *m_t_*_1_ is the total mass of the water, jar, and glass sheet (g); *m_t_*_2_ denotes the total mass for the aggregate, water, jar, and glass sheet (g); and *ρ_w_* = 1000 kg/m^3^, which means the water density.

#### 2.4.3. Water Absorption

The water absorption of the RCA samples before and after carbonation was measured according to the method specified in GB/T14685-2022. The expression for the calculation of water absorption is shown in Equation (3):(3)$$W=\frac{m_{s}-m_{d}}{m_{d}}\times 100\%$$
where *W* is the water absorption of the aggregate (%), *m_s_* is the mass of the saturated surface dry aggregate (g), and *m_d_* means the mass of aggregate after completely drying (g).

#### 2.4.4. Actual Mass Increase Rate

To prevent moisture inside the RCA sample from influencing the mass gain of the sample after complete carbonation, the actual mass increase rate was used to represent the proportion of the solid carbonation reaction product of the RCA sample after carbonation in the total solid amount of the sample. The mass of water in the sample was excluded in the calculation of the actual mass increase rate. Therefore, the actual mass increase rate could be considered as a normalized value to reflect the carbonate sequestration degree of RCA samples. The greater the actual mass increase rate was, the more products were produced by the carbonation reaction, and the more completely CO_2_ was absorbed by the RCA sample. The expression for the calculation of the rate of actual mass increase is shown in Equation (4):(4)$$\Delta m_{ar}=\frac{\Delta M_{e}}{M_{0}(1-IWC)}\times 100\%$$
(5)$$\Delta M_{e}=M_{h}-M_{0}(1-\;\mathrm{IWC}\;)$$
where Δ*m_ar_* is the rate of actual mass increase in the RCA (%), Δ*M_e_* is the actual mass increase for the RCA sample after CO_2_ carbonation treatment (calculated by Equation (5)) (g), *M*_0_ is the mass of the RCA before carbonation (g), the IWC is the initial water content before the carbonation of the RCA (see Table 2 for specific values) (%), and *M_h_* is the mass of the RCA after CO_2_ carbonation treatment and completely drying (g).

#### 2.4.5. Carbonation Ratio

The carbonation ratio was used to quantify the carbonation degree of the RCA sample. The larger the carbonation ratio was, the greater the carbonation degree of the RCA sample. The expression for the calculation of the carbonation ratio is shown in Equation (6):(6)$$\varepsilon =\frac{\Delta M_{e}}{\Delta M_{t}}\times 100\%$$
where *ε* is the carbonation ratio of the RCA (%) and Δ*M_e_* is the actual mass increase in the RCA (calculated using Equation (5)) (g). Δ*M_t_* is the theoretical mass increase for the RCA sample, which represents the maximum theoretical absorption of CO_2_ by cement mortar adhered to the surface of recycled aggregate. The expression for the calculation of Δ*M_t_* is shown in Equation (7) [38]:(7)$$\Delta M_{t}=\frac{M_{c}\cdot CO_{2}{\%}_{max}}{\left(M_{c}+M_{s}+M_{a}+0.23\cdot M_{c}\right)(1+\;\mathrm{IWC}\;)}$$
where *M_c_*, *M_s_*, and *M_a_* are the proportions of cement, sand, and coarse aggregate in the original concrete, respectively (calculated from Table 1) (%). The IWC is the initial water content before the carbonation of the RCA (see Table 2 for specific values) (%). *CO*_2_%_max_ is the maximum theoretical amount of CO_2_ captured by Portland cement, which was calculated from the oxide content of cement [43]; according to the literature [44], *CO*_2_%_max_ = 50%.

## 3. Results

### 3.1. Water Content

Figure 3 shows the measured results of the water content of the RCA samples before and after carbonation under different moisture conditions.

The water contents of the dry RCA samples after carbonation were larger than before carbonation, and the water contents of the samples after carbonation increased with increasing RH. The water contents of the saturated RCA samples were smaller after carbonation than before carbonation, and the water contents of the samples after carbonation increased with increasing RH. The water contents of natural RCA samples after carbonation were smaller than before carbonation in the carbonation environment where RH = 50% and 70%, but larger than before carbonation in the carbonation environment where RH = 90%.

(1)The water contents of the dry RCA samples after carbonation were greater in the three different RH carbonation environments than before carbonation, and the water contents of the samples after carbonation increased with increasing RH. The results showed that the direction of moisture transport in the dry RCA samples during the carbonation reaction was from the carbonation environment to the interior of the RCA samples. Additionally, the higher the RH was, the higher the amount of moisture that penetrated into the interior of the RCA. The moisture that infiltrated the interior of the sample could provide conditions for the carbonation reaction of the dry RCA samples.(2)The water contents of the natural RCA samples after carbonation were smaller in the carbonation environments where RH = 50% and 70% than before carbonation, while the water contents after carbonation were greater in the carbonation environment where RH = 90% than before carbonation. The results showed that during the carbonation process, the direction of moisture transport between the natural-state RCA samples and the carbonation environments with RH ≤ 70% and RH = 90% was reversed. When RH ≤ 70%, moisture transferred from the RCA interior to the carbonation environment. When RH = 90%, moisture transferred from the carbonation environment to the RCA samples. Thus, moisture transport between the natural RCA samples and the carbonation environment would reach a state of almost stagnation at a certain relative humidity between RH = 70% and RH = 90%.(3)The water contents of the saturated RCA samples after carbonation were lower in three different RH carbonation environments than before carbonation, and the water contents of the samples after carbonation increased with increasing RH. The results showed that the direction of moisture transport in the saturated RCA samples during the carbonation reaction was outward leakage from the RCA interior to the carbonation environment. The moisture leakage from the RCA interior to the carbonation environment decreased with increasing RH, which meant that much moisture remained in the samples when saturated RCA samples were carbonized in the environment with higher RH. Based on the measured results of other properties, this moisture had a certain impact on the effect of carbonation on the modification of RCA samples; the specific analysis is discussed below.

### 3.2. Actual Mass Increase Rate

Figure 4 shows the measured results of the actual mass increase rate of the CRCA samples under different moisture conditions, as calculated using Equations (4) and (5).

The actual mass increase rates of the dry RCA samples after carbonation in the RH = 70% and 90% carbonation environments were higher than that in the RH = 50% carbonation environment. The actual mass increase rate of the natural RCA sample after carbonation was the lowest among the three different IWC samples. The saturated RCA sample had a large actual mass increase rate after carbonation in three different RH carbonation environments, and the actual mass increase rate decreased with increasing RH.

(1)The dry RCA sample had a higher actual mass increase rate after carbonation in the carbonation environment with higher RH, especially in the carbonation environment with RH = 70%, and the actual mass increase rate reached 2.38%, which was one of the several groups of specimens with excellent performance in this experiment. The results showed that the larger RH in the carbonation environment promoted the absorption of CO_2_ by the dry RCA sample during the carbonation reaction. The reason for this was that the CO_2_ gas was more soluble in the carbonation environment with higher RH and diffused in the pores of RCA as the moisture penetrated into the interior. In addition, the larger RH was conducive to the infiltration of moisture from the environment into the dry RCA samples, which promoted the absorption of CO_2_ by the dry RCA sample.(2)The actual mass increase rate of the natural RCA sample after carbonation in three different RH carbonation environments was small, and the maximum was only 1.55%. The actual mass increase rate of the natural RCA sample in the RH = 90% carbonation environment was 0.92%, which was the minimum actual mass increase rate in this experiment. In addition, the actual mass increase rate of the natural RCA sample decreased with increasing RH. A comprehensive comparison of samples with the three different IWCs showed that the actual mass increase rate of the natural RCA sample after carbonation was the lowest.(3)The saturated RCA sample had an excellent actual mass increase rate after carbonation in the three different RH carbonation environments. The actual mass increase rate of the saturated RCA sample decreased with increasing RH. The actual mass increase rates of the saturated RCA samples reached 2.48% and 2.40%, respectively, in the carbonation environments with RH = 50% and 70%, respectively, which were the two groups with the best actual mass increase rates in this experiment. According to the analysis, the higher IWC of the saturated RCA sample increased the solubility of calcium ions in the internal hydration products of RCA. After CO_2_ penetrated the sample, it directly reacted with the highly soluble calcium ions, and thus, the saturated RCA sample absorbed CO_2_ more completely.

### 3.3. Carbonation Ratio

Figure 5 shows the measured results for the carbonation ratios of the RCA samples after carbonation under different moisture conditions, as calculated using using Equations (5)–(7):

In the carbonation environment with RH = 50%, the carbonation ratio of the RCA sample increased with increasing IWC and reached 30.89% when the RCA sample was saturated; this was the largest carbonation ratio group in this experiment. In the carbonation environments with RH = 70% and 90%, the carbonation ratios of the dry and saturated RCA samples were higher than those of the natural RCA samples.

(1)The carbonation ratio of the dry RCA sample reached 28.01% under the carbonation environment of RH = 70%. In the carbonation environment of RH = 90%, the carbonation ratio was 21.65%, and the carbonation rate of the dry RCA sample was the smallest when RH = 50%. Since the IWC of the dry RCA sample was very small and even negligible, the moisture conditions required for the carbonation reaction of the sample were all dependent on the environmental RH. The results showed that different environmental RHs had a great influence on the carbonation ratio of dry RCA samples, and the environmental RH should not be too high or too low.(2)The carbonation ratio of the natural RCA sample in the carbonation environments where RH = 50% and RH = 70% was higher than that where RH = 90%. Combined with the measured results of the water content of the natural RCA sample before and after carbonation, the results showed that in the carbonation environments with RH = 50% and RH = 70%, the moisture in the natural RCA sample during the carbonation process permeated from inside the RCA to the carbonation environment. This reduced the water content in the sample and reduced the hindrance of diffusion of CO_2_ in the pores of the sample. In the carbonation environment of RH = 90%, the moisture in the natural RCA sample was transferred from the carbonation environment to the interior of the RCA sample during the carbonation process, which increased the water content in the sample and the hindrance of the diffusion of CO_2_ in the pores of the sample. Therefore, the carbonation ratio of the natural RCA sample in the RH = 50% and RH = 70% carbonation environments was larger than that for RH = 90%.(3)The carbonation ratios of the saturated RCA samples after carbonation in three different RH carbonation environments were greater than those of the dry and natural RCA samples. The carbonation ratios of the saturated RCA samples reached 30.89% and 29.92% in the carbonation environments where RH = 50% and RH = 70%, respectively; these were the two groups with the largest carbonation ratios in this experiment. Due to the large IWC of the saturated RCA sample, calcium ions were more soluble, and the calcium ions that dissolved after CO_2_ infiltration into the internal pores of the sample reacted directly, and thus, the carbonation ratio of the saturated RCA sample was larger than that of the dry and natural RCA samples. In the environment of RH = 90%, the water inside the saturated RCA sample was less likely to seep out than that of RH = 50% and RH = 70%. The residual water hindered the diffusion of CO_2_ inside the sample, resulting in the carbonation ratio of the saturated RCA sample in the environment of RH = 90% being less than that of RH = 50% and RH = 70%.

### 3.4. Apparent Density

Figure 6 shows the measured apparent density for the CRCA samples under different moisture conditions as calculated by Equation (2):

In the carbonation environment with RH = 50%, the enhancement effect of the apparent density for the saturated RCA sample after carbonation was greater than that of the dry and natural samples. The carbonation environment with RH = 70% had the best effect on improving the apparent density of the RCA, and the apparent density of the CRCA decreased with increasing IWC. In the carbonation environment with RH = 90%, the apparent density of the dry RCA sample was the best.

(1)In the carbonation environment with RH = 50%, the apparent density of the saturated RCA sample improved after carbonation, and the apparent density was 3.19% larger than that before carbonation. The improvements of the apparent densities of the dry and natural RCA samples were similar for the carbonation environment of RH = 50%, and the apparent densities were 1.6~1.7% greater than before carbonation. Combined with the measured results of the water content, these results showed that the water contents of the RCA-NS-50 and RCA-SS-50 samples after carbonation were very similar (see Figure 3), but the difference in the apparent density improvement between the two groups was very obvious. According to the analysis, the reason for this was that the RCA-SS-50 sample had a large IWC before carbonation, which increased the solubility of calcium ions. Meanwhile, the sample with a larger IWC enhanced the ability of CO_2_ to permeate RCA and promoted the carbonation reaction. Therefore, the improvement in the apparent density of the RCA-SS-50 sample was greater than that of the RCA-NS-50 sample.(2)Comprehensively comparing the measured results of the apparent density of the RCA samples after carbonation in the three different RH carbonation environments showed that the carbonation environment of RH = 70% had an excellent improvement effect on the apparent density of the RCA samples. The apparent density of the dry RCA sample increased by 3.95% after carbonation, which was the group that showed the greatest improvement in the apparent density in this experiment. According to the measured results for the apparent density, it was not difficult to find that in the carbonation environment with RH = 70%, the apparent density of CRCA decreased with increasing IWC, which indicated that the IWC had a certain influence on the improvement in the apparent density. The larger IWC inhibited the diffusion of CO_2_ in the samples, which limited the increase in the apparent density.(3)The carbonation environment with RH = 90% had a very small improvement effect on the apparent density of the RCA samples. The apparent density of the dry RCA sample increased by 1.77% in the carbonation environment where RH = 90%. The apparent densities of the natural and saturated RCA samples increased by 0.61% and 0.48%, respectively, in the carbonation environment with RH = 90%. Combined with the measured results of water content (see Figure 3), these results showed that in the carbonation environment with RH = 90%, moisture was transferred from the carbonation environment to the interior of the natural RCA sample during the carbonation process. Less moisture seeped from inside the saturated RCA sample to the carbonation environment during the carbonation process. This meant that there was still a large amount of moisture inside the sample when the natural and saturated RCA samples were carbonized in the carbonation environment with RH = 90%. Excessive moisture increased the barrier to CO_2_ diffusion in the pores of the sample, which resulted in a very small enhancement in the apparent density of the sample.

### 3.5. Water Absorption

Figure 7 shows the measured results of the water absorption of the RCA samples after carbonation under different moisture conditions, as calculated using Equation (3).

In the carbonation environment where RH = 50%, the improvement effect of water absorption for the saturated RCA sample was better than that of the dry and natural RCA samples. In the carbonation environment of RH = 70% and 90%, the improvement effect of water absorption on the dry RCA sample was better than that of the natural and saturated RCA samples.

(1)In the carbonation environment where RH = 50%, the water absorption of the saturated RCA sample after carbonation was significantly improved, and the water absorption was reduced by 1.41% compared with that before carbonation, which was the best group that displayed the best improvement effect from water absorption in this experiment. The dry and natural RCA samples showed similar improvements from water absorption in the carbonation environment where RH = 50%, and the effect was reduced by 0.77% and 0.76%, respectively, compared with that before carbonation. For the carbonation environment where RH = 50%, the trend for the improvement related to water absorption in the three different IWC samples was the same as that of the improvement effect for apparent density. Combined with the measured results of the actual mass increase rate and carbonation ratio of different samples, these results showed that the properties of the saturated RCA sample had the best overall improvement effect in the carbonation environment where RH = 50%.(2)In the carbonation environment where RH = 70%, the improvement effect of water absorption for the dry RCA sample after carbonation was slightly better than that for the natural and saturated RCA samples, and the water absorption was 1.16% less compared with that before carbonation. The natural and saturated RCA samples had similar water absorption improvement effects in the carbonation environment where RH = 70%, and the water absorption was reduced by 0.98% and 0.95%, respectively, compared with that before carbonation. The results showed that there was little difference in the water absorption improvement effects for samples with the three different IWCs in the carbonation environment with RH = 70%. Therefore, for the property index of water absorption, the improvement related to water absorption by samples in the carbonation environment of RH = 70% were less affected by the IWC.(3)In the carbonation environment where RH = 90%, the water absorption of the dry RCA sample after carbonation was reduced by 1.32% compared with that before carbonation. When the natural and saturated RCA samples were carbonized in the carbonation environment where RH = 90%, there was still a large amount of moisture in the sample, which hindered the diffusion of CO_2_ in the pores and resulted in the improvement in the water absorption of the natural and saturated RCA samples to less than that of the dry RCA sample. However, in the carbonation environment of RH = 90%, the water absorption improvement effect of the saturated RCA sample was better than that of the natural RCA sample. This may have been due to the higher IWC of the saturated RCA sample, the greater solubility of calcium ions in the hydration products of the sample, and the more favorable carbonation reaction with CO_2_.

## 4. Discussion

According to this experimental study, the carbonation environmental RH and the IWC of RCA were the main water sources in the carbonation reaction of RCA samples, and the moisture conditions created by the two sources had a very obvious effect on the improvement in CRCA properties (including the actual mass increase rate, carbonation ratio, apparent density, and water absorption). The synergistic effect of carbonation environmental RH and IWC on the improvement in the properties of CRCA is discussed below.

When the RCA sample was under the condition of high RH with low IWC (as shown in Figure 8a), due to the low initial water content of the RCA sample, the moisture of the carbonation environment easily penetrated into the RCA sample and diffused rapidly in the pores. In addition, due to the high RH of the carbonation environment, CO_2_ gas was very soluble in the environment and diffused in the pores of RCA after moisture penetrated the interior of the RCA sample. When the RCA sample was carbonized under such moisture conditions, CO_2_ had excellent abilities to permeate and diffuse, which improved the properties of the CRCA samples, such as the RCA-DS-70 and RCA-DS-90 samples in this experiment. Figure 9a,b show the microstructures of the original RCA sample before carbonation and the RCA-DS-70 sample after carbonation, respectively. The ITZ was filled very well with carbonation products after carbonation. In addition, the environmental RH should not have been too high. According to the measured results of the actual mass increase rate, carbonation ratio, and apparent density of the RCA-DS-70 and RCA-DS-90 samples, the overall improvement in CRCA properties was better in the environment where RH = 70% than RH = 90%.

When the RCA sample was under the condition of low RH with high IWC (as shown in Figure 8b), due to the high initial water content of the RCA sample, the ability of CO_2_ gas to permeate increased, and more CO_2_ penetrated the internal pores of the RCA solid. In addition, due to the low RH of the carbonation environment, the direction of moisture transport during the carbonation reaction was from the inside of the RCA sample to the environment, which solved the problem of limited CO_2_ diffusion due to high moisture content inside the RCA to a certain extent. The ability of CO_2_ gas to permeate was better, and the diffusion process was less hindered when the RCA sample was carbonized under this moisture condition. Thus, there was a good effect on the improvement in the properties of the CRCA sample. For example, for this paper’s experimental study, the water content of the RCA-SS-50 sample was 5.88% before carbonation and 1.82% after carbonation, which indicated that a large amount of moisture inside the sample flowed into the environment during the carbonation reaction process. Additionally, according to the measured results of the properties of the RCA-SS-50 sample, the overall improvement in the properties of the sample after carbonation was excellent. Figure 9a,c show the microstructures of the original RCA sample before carbonation and the RCA-SS-50 sample after carbonation, respectively. The ITZ was filled very well with carbonation products after carbonation.

When the RCA sample was under the condition of high RH with high IWC (as shown in Figure 8c), although the high initial water content of the RCA sample increased the ability of CO_2_ to permeate the samples, the high RH of the carbonation environment limited the leakage of moisture from the inside of the RCA sample to the environment. This resulted in there still being excessive moisture in the internal pores of the RCA sample, which limited the penetration and diffusion of CO_2_ in the internal pores of the RCA. For example, in this experiment, the water contents of the RCA-SS-70 and RCA-SS-90 samples after carbonation were 2.40% and 3.09%, respectively. In addition, the two groups of samples had higher actual mass increase rates and carbonation ratios, but the improvement effects of apparent density and water absorption were very small. This may have been because the excessive moisture in the samples hindered the diffusion of CO_2_, resulting in only the local carbonation of cement mortar in the surface area of the RCA; the deeper pore structure was not effectively improved. Figure 9a,d show the microstructures of the original RCA sample before carbonation and the RCA-SS-90 sample after carbonation, respectively; the ITZ was generally filled with carbonation products after carbonation.

When the RCA sample was under the condition of low RH with low IWC (as shown in Figure 8d), due to the low initial water content of the RCA sample, the ability of CO_2_ to permeate RCA was poor. Additionally, due to the low RH of the carbonation environment, the infiltration of the RCA sample by moisture from the environment was limited, which resulted in the insufficient ability of CO_2_ to diffuse in the internal pores of the RCA sample. The abilities of CO_2_ gas to permeate RCA and diffuse were poor when the RCA sample was carbonized under such moisture conditions. For example, in this experiment, the water content of the RCA-DS-50 sample after carbonation increased to only 1.19%, and less moisture from the environment infiltrated the sample, which inhibited the penetration and diffusion of CO_2_ and resulted in a little improvement in the properties of the sample. Figure 9a,e show the microstructure of the original RCA sample before carbonation and the RCA-DS-50 sample after carbonation; the ITZ was generally filled with carbonation products after carbonation.

There was also a special moisture condition where RH and IWC were in a state of relative equilibrium, or the moisture gradient between the RH and IWC of the RCA sample was small, such as the RCA-NS-90 sample in this experiment. According to the measured results of the properties, the RCA-NS-90 sample was the worst in this experiment. The analysis shows that when the RCA sample was carbonized under the moisture condition where RH and IWC were in a state of relative equilibrium (as shown in Figure 8e), moisture transport between the interior of the RCA sample and the carbonation environment was almost stagnant or very slow, which was unfavorable for CO_2_ to penetrate and diffuse in the sample and resulted in very small improvements in the properties of the sample. Figure 9a,f show the microstructure of the original RCA sample before carbonation and the RCA-NS-90 sample after carbonation; the ITZ was less filled with carbonation products after carbonation than in other cases.

The above content discusses the transport of water and CO_2_ during the carbonation reaction of RCA samples under five representative moisture conditions. The effect of carbonation on the performance improvement of RCA samples under different moisture conditions was analyzed. In the published literature, research on the effect of moisture on the carbonation modification of RCA focused on either environmental RH or the IWC of RCA, but not both. For example, in the studies of Fang et al. [44], the carbonation environment of RCA was set to different relative humidities (RH = 5%, 50%, and 95%), but the initial water content of RCA was unknown. In the investigations of Zhan et al. [38], RCA samples with different IWCs (IWC = 0.08%, 3.37%, and 5.03%) were prepared, but the relative humidity of the environment was not a fixed value during the carbonation process, but fluctuated between 40% and 70%. Hence, for this paper’s studies, the carbonation tests of RCA samples with different IWCs under different constant RHs were carried out to deeply explore the synergistic effect of the environmental RH and IWC on RCA carbonation modification and to further compensate for the deficiencies of the previous efforts in the published literature.

## 5. Conclusions

The experimental study of the accelerated carbonation of RCAs with different IWC states in different RH carbonation environments showed that the combined effects of IWC and RH had a significant effect on the improvement in the properties of CRCA. The different moisture conditions created by the IWC and RH affected the property improvement in RCA after carbonation by influencing the abilities of CO_2_ to penetrate and diffuse in CRCA. The specific conclusions are as follows:(1)The RCA sample had a higher actual mass increase rate and carbonation ratio after carbonation under the moisture conditions of RH = 70% for the dry-state IWC (RCA-DS-70) and RH = 50% for the saturated-state IWC (RCA-SS-50). The apparent density and water absorption of RCA samples also had relatively excellent effects on the improvements in properties.(2)The saturated RCA sample after carbonation treatment had a large carbonation ratio and actual mass increase rate, but the apparent density and water absorption had very small effects on the improvement in properties. This may have been because it was difficult for CO_2_ gas to penetrate deep inside the saturated RCA sample, and the deeper pore structure was not effectively improved.(3)The enhancement effect of the apparent density of the RCA sample was the best in the carbonation environment where RH = 70%, and the improvement in the apparent density decreased with increasing IWC. In the carbonation environment where RH = 50%, the sample with the best water absorption improvement effect was the saturated RCA sample. In the carbonation environments where RH = 70% and RH = 90%, the samples with the best water absorption improvement effect were dry RCA samples.(4)Under the moisture conditions of high RH with low IWC and low RH with high IWC, CO_2_ had good abilities to permeate RCA and diffuse, and the moisture conditions caused excellent overall improvements in the properties of RCA after carbonation. Under the moisture conditions of high RH with high IWC, low RH with low IWC, and RH and IWC in a state of relative equilibrium, the abilities of CO_2_ to penetrate and diffuse were hindered to varying degrees, and the moisture conditions had small effects on the improvement in the properties of RCA after carbonation.

## Figures and Tables

**Figure 1 materials-16-05251-f001:**
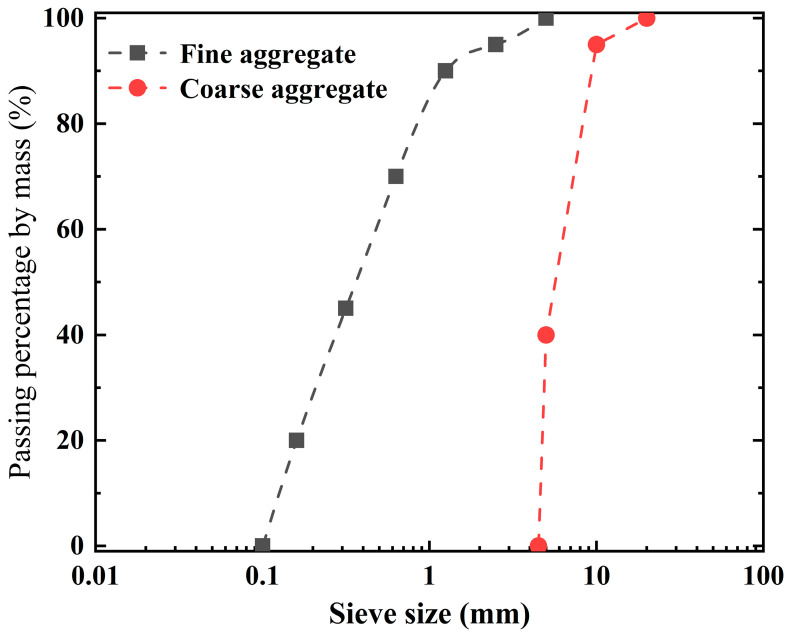
Particle size distribution of fine and coarse aggregate.

**Figure 2 materials-16-05251-f002:**
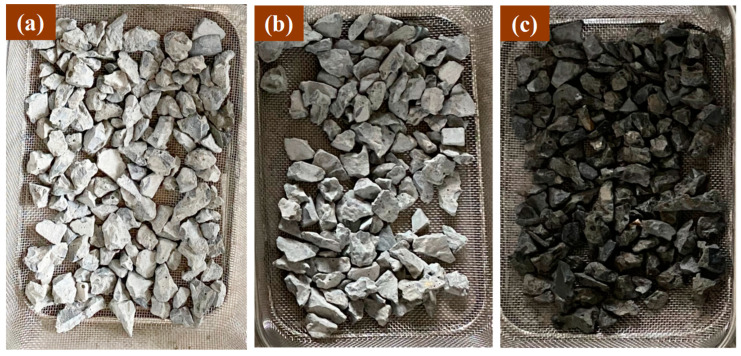
RCA samples with different IWCs after pretreatment: (**a**) dry state (IWC = 0%); (**b**) natural state (IWC = 2.66%); (**c**) saturated state (IWC = 5.88%).

**Figure 3 materials-16-05251-f003:**
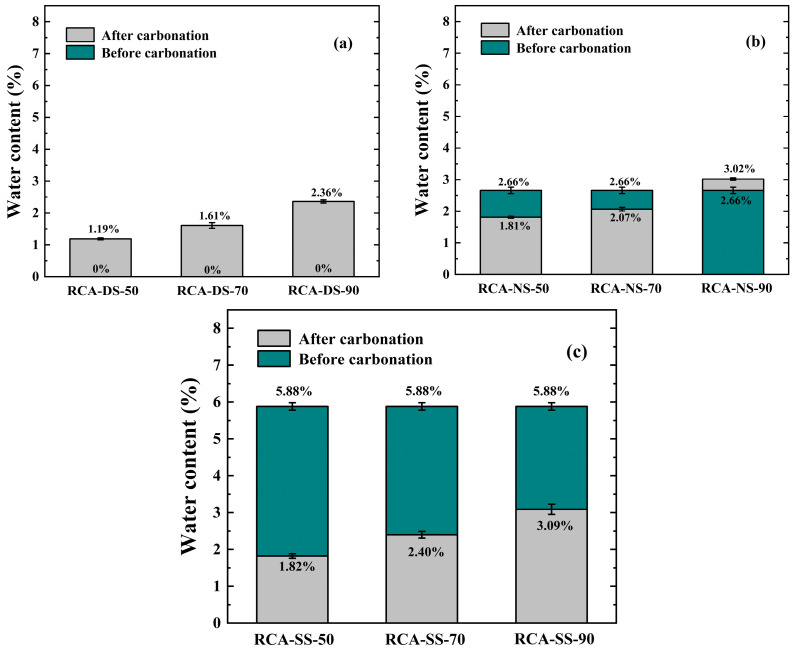
Water contents of RCA samples before and after carbonation: (**a**) dry state (IWC = 0%); (**b**) natural state (IWC = 2.66%); (**c**) saturated state (IWC = 5.88%).

**Figure 4 materials-16-05251-f004:**
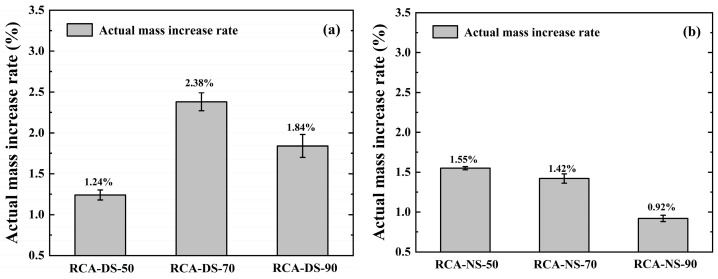
Actual mass increase rates of RCA: (**a**) dry state (IWC = 0%); (**b**) natural state (IWC = 2.66%); (**c**) saturated state (IWC = 5.88%).

**Figure 5 materials-16-05251-f005:**
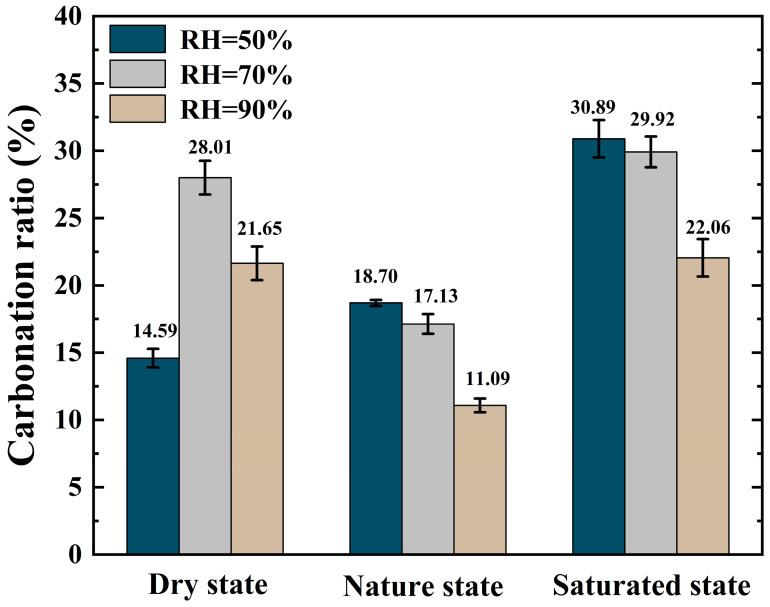
Carbonation ratios of RCA.

**Figure 6 materials-16-05251-f006:**
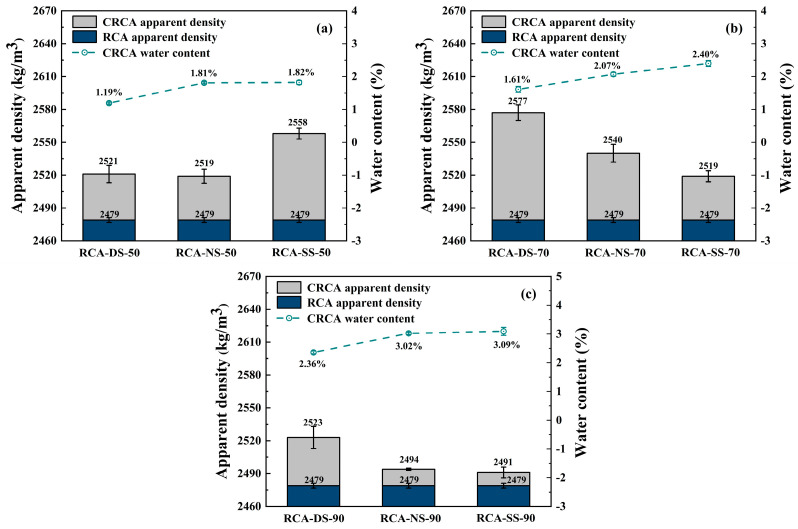
Apparent density of RCA before and after carbonation: (**a**) RH = 50%; (**b**) RH = 70%; (**c**) RH = 90%.

**Figure 7 materials-16-05251-f007:**
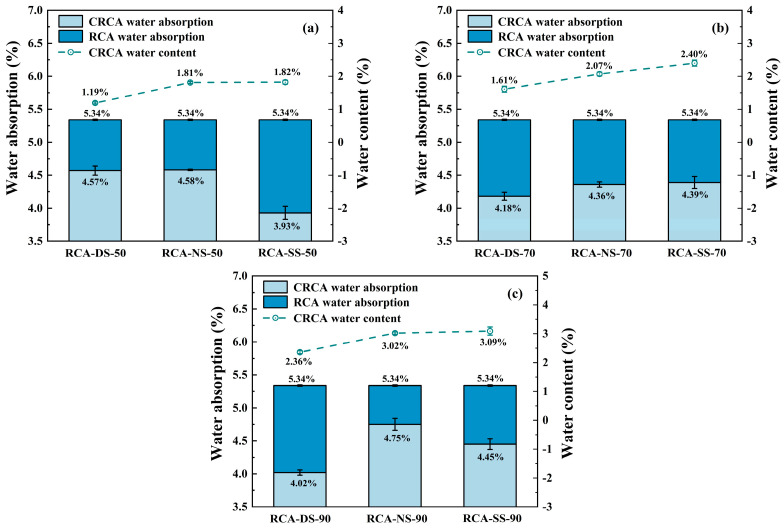
Water absorption of RCA before and after carbonation: (**a**) RH = 50%; (**b**) RH = 70%; (**c**) RH = 90%.

**Figure 8 materials-16-05251-f008:**
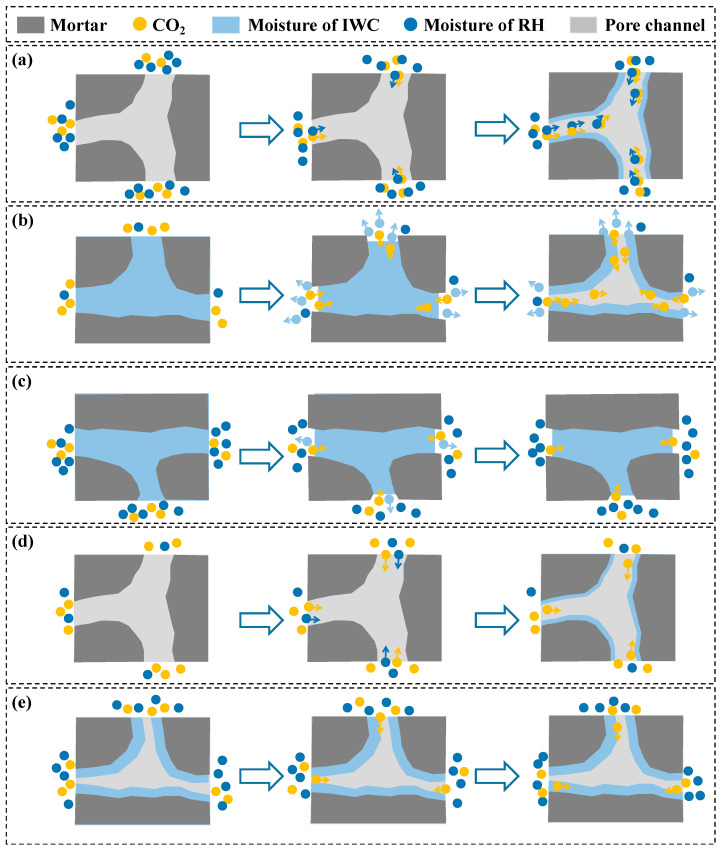
Different moisture conditions in carbonation process of RCA: (**a**) high RH with low IWC; (**b**) low RH with high IWC; (**c**) high RH with high IWC; (**d**) low RH with low IWC; (**e**) RH and IWC in relative equilibrium.

**Figure 9 materials-16-05251-f009:**
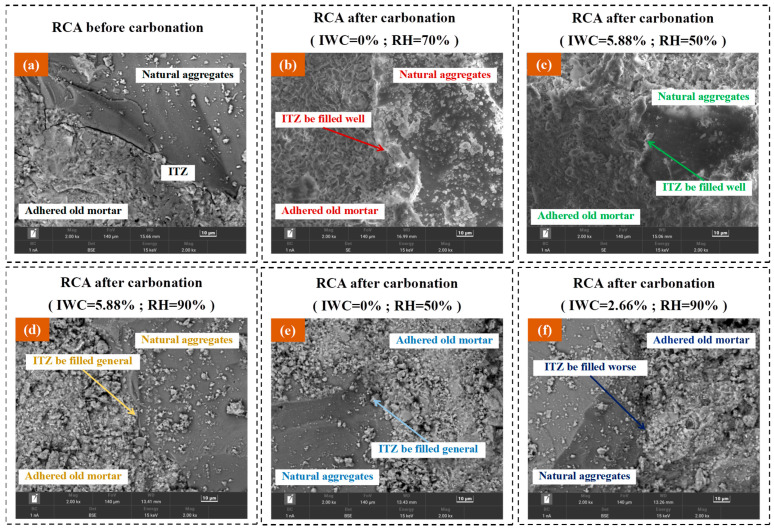
SEM images of RCA and CRCA: (**a**) original RCA sample; (**b**) RCA-DS-70 sample after carbonation; (**c**) RCA-SS-50 sample after carbonation; (**d**) RCA-SS-90 sample after carbonation; (**e**) RCA-DS-50 sample after carbonation; (**f**) RCA-NS-90 sample after carbonation.

**Table 1 materials-16-05251-t001:** Mix proportions of the original concrete.

w/c	Water (kg/m^3^)	Cement (kg/m^3^)	Fine Aggregate (kg/m^3^)	Coarse Aggregate (kg/m^3^)	28 d Compressive Strength (MPa)
0.50	195.0	390.0	617.1	1197.9	41.6

**Table 2 materials-16-05251-t002:** Different curing conditions for RCA samples.

Samples Type	Temperature (℃)	CO_2_ Concentration (%)	RH (%)	IWC State	IWC (%)
RCA-DS-50	20	20	50	Dry state	0.00
RCA-DS-70			70		
RCA-DS-90			90		
RCA-NS-50	20	20	50	Natural state	2.66
RCA-NS-70			70		
RCA-NS-90			90		
RCA-SS-50	20	20	50	Saturated state	5.88
RCA-SS-70			70		
RCA-SS-90			90		

## Data Availability

Not applicable.

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
