# Peer review of "Synergistic Effects of Environmental Relative Humidity and Initial Water Content of Recycled Concrete Aggregate on the Improvement in Properties via Carbonation Reactions"

_materials, 2023, doi:10.3390/ma16155251_

Round 1

Reviewer 1 Report

Manuscript ID: materials-2502744

General comments:

English proofing of the document is required there are abundant grammatical and structural mistakes with the sentences presented in the paper. Often the order of arguments presented in each section are incorrect which has resulted in mixed messaged being given to readers.

Specific comments:

Line 57-58: Very important papers in the field have covered and discussed chemical equations presented in this section. Those papers have not been acknowledged in here. I suggest authors refer to the papers and cite in this section. Examples of the papers are provided in here:

Manning, D. A. C.; Renforth, P. Passive sequestration of atmospheric CO2 through coupled plant-mineral reactions in urban soils. Environmental Science & Technology 2013, 47, 135–141.

Jorat, M.E., Kraavi, K.E., Manning, D.A.C. (2022). Removal of atmospheric CO2 by engineered soils in infrastructure projects. Journal of Environmental Management. 314, 115016.

Jorat, M.E., Goddard, M.A., Manning, P., Lau, H.K., Ngeow, S., Sohi, S.P., Manning, D.A.C. (2020). Passive CO2 removal in urban soils: Evidence from brownfield sites. Science of the Total Environment, 703, 135573.

Washbourne, C. L.; Lopez-Capel, E.; Renforth, P.; Ascough, P. L.; Manning, D. A. C. Rapid removal of atmospheric CO2 by urban soils. Environmental Science & Technology 2015, 49 (9), 5434-5440.

Jo, H. K.; McPherson, G. E. Carbon storage and flux in urban residential greenspace. Journal of Environmental Management 1995, 45 (2), 109-133.

Line 194: Is there a reference for this equation? It is always tricky to use mass balance for calculation of carbonates.

Line 217: This comment is not only specific to this sentence but have been repeated throughout the paper. I could not understand the term “The water contents of the dry RCA samples”. If a RCA sample is dried, then the water content should be zero, how can the sample then have a value for the moisture content. There must be a reason for this but the authors have not made it clear.

Line 251: This comment is for the whole of this section. In relation to mass gain, I suggest normalising mass to take away mass of the water. If you compare absolute mass, then you are taking into account mass of water in addition to mass of aggregates and seeing how aggregates carbonate sequestration varies.

Line 333: It is not clear to me why apparent density is an important element related to this study and needed to be investigated and presented in the paper.

Line 431: The discussion section is strong in relation to discussing results from this study. I think the authors did a good job at providing and in-depth discussion on what their data shows but the discussions section lacks comparison of data to published literature. How does your results fit the current literature. Has different your results are compared to the papers published in the field? In your research, you mentioned “In summary, recent research on the effect of moisture on the carbonation modification of RCA focused on either environmental RH or the IWC of RCA, but not both.” So there are studies in the field and the authors need to properly discuss their results compared with the published literature.

Line 435: Mention what type of properties you are speaking about.

Must be improved, there are substantial mistakes structurally and grammatically.  

Reviewer 2 Report

The subject of the authors' research is the effects of environmental relative humidity and initial water content of recycled concrete aggregate on the improof of properties by carbonation reactions.

The experimental study of the accelerated carbonation of recycled concrete aggregates with different initial water content states in different relative humidity carbonation environments showed that the combined effects of initial water content and relative humidity had a significant effect on the improvement of the properties of carbonated recycled concrete aggregate .

Manuscript evaluation

Positive Remarks:

1. Recycled concrete aggregates introduced to production more and more often require information about the durability of such concrete.

2. The authors clearly defined the scope of the experiment and adopted two, also in my opinion, the most important parameters:

- environmental relative humidity

- the initial water content of the recycled concrete aggregate itself.

Too many variable parameters often impair the readability of the results and do not allow for statistical analysis.

Critical remarks:

1. There is no information on the structure of fine and coarse aggregate (aggregate screening curve, information on fractions)

2. It would be advisable to provide recipe of the concrete  before recycling. In particular, the type of cement and the physical characteristics of the aggregate.

Overall, the article is correct and should be published.

Reviewer 3 Report

The paper titled “Synergistic effects of environmental relative humidity and initial water content of recycled concrete aggregate on the improvement of properties by carbonation reactions” is a great paper to be published in the materials journal, however, some change may need to be done by authors to increase the strength of the paper.

Abstract

Some results at the end of this paragraph are great to show with comparison.

Keywords: it is okay

Introduction

Many studies are missing such as recycling sand blocks, and concrete block, please read more papers and make better literature such as “An Environmental Sustainability Roadmap for Partially Substituting Agricultural Waste for Sand in Cement Blocks”.

Line 74: please take out each cited paper and explain them from 29 to 34 or remove some of them if you do not.

Experiment and methods

Figure 2 : I feel it is unnecessary to show, authors could show better pictures and schematic illustration, here. Please revise.

Table 2: why is name scope of experiments? It is name of sample and tests? Please explain or revise.

Results

Figure 3 is that result exact result without adding the values for the standard deviation, so how you confirm only one sample is enough.

Figure 4 what is that link between all three water content to actual mass things

Figure 5 what is the reason with high RH in saturated state, you got a lower carbonation rate compare the two others. Please explain.

Figure 8 is that figure true for the all types of the mortar, please give an explain.

Conclusion

Overall this part is okay

References

May add some literature will help to increase the importance of the paper. 

Overall, the writing is okay. 

Round 2

Reviewer 1 Report

Revisions made to the paper have improved the quality significantly. I suggest authors to grammatically check the text

Revisions made to the paper have improved the quality significantly. I suggest authors to grammatically check the text

Reviewer 3 Report

Now, the paper is good at the format

It is okay, in regard to English language but it can be better.